# Preliminary Analysis of the Water Quality Status in an Urban Mediterranean River

Christina Papadaki *, Sergios Lagogiannis and Elias Dimitriou

Hellenic Centre for Marine Research, Institute of Marine Biological Resources and Inland Waters,
46.7 km of Athens-Sounio Ave., 19013 Athens, Greece; elias@hcmr.gr (E.D.)
* Correspondence: chrispap@hcmr.gr; Tel.: +30-22910-76349

**Abstract:** Recreational use of urban rivers is becoming popular since rivers may act as amenity corridors with the potential for exhibition, recreation, leisure, relaxation and retreat. However, several point and non-point pollution sources contribute to the degradation of urban rivers' water quality, thereby impeding their beneficial uses and amenities. The physicochemical and microbiological quality of a Greek urban river (Kifisos–Athens) was analyzed over a period of 12 months. A sampling campaign was implemented, collecting monthly data from five sites. Spearman's analysis showed significant correlation of the Hellenic Water Quality Index with specific nutrients. The total physicochemical status of all sampling stations was characterized as poor or bad. The annual average concentration of *Escherichia coli* (*E. coli*) was extremely high in four sites out of five, ranging from 16,822 to 26,780 cfu/100 mL. Bacteriological quality was unacceptable, as the study demonstrated the widespread occurrence of *E. coli* and low-quality physiochemical conditions. The spatiotemporal distribution of pollution levels revealed hotspots to be monitored further via automatic monitoring stations. A series of management and restoration measures, including tracing the exact pollution routes, should be initiated to minimize pollution pressures and establish the good ecological status of an important Mediterranean river.

**Keywords:** urban river; recreational; water quality; *Escherichia coli*; statistical analysis





## 1. Introduction

Urban rivers are important ecosystems that promote health, prosperity and wellbeing in cities, regarding both public welfare and community livelihood. Rivers offer places to walk next to river banks, boat or paddle, fish, or just sit and relax, refresh, and renew. It is therefore crucial that people are able to safely access and interact with these spaces. Nevertheless, several pollution sources driven by human activity contribute to urban river pollution, including direct discharge from industrial facilities and sewage treatment systems, agricultural runoff, recreational activities themselves and illegal actions [1]. Urban diffuse pollution from cities and transport carries pollutants from roads and urban surfaces into storm drains and out into watercourses at many locations along the river. Moreover, the atmospheric deposition, storm water and urban runoff from urban surfaces may carry various contaminants [2] including toxic metals, organic pollutants, pathogenic organisms and pharmaceuticals such as antibiotics, which create extensive environmental and human health concerns [3]. Additionally, many urban rivers have been channeled, buried or otherwise confined with the impacts of hydro-morphological alterations, enhancing the risk of further degradation in both water quality and quantity [4,5].

Water quality conditions are commonly assessed by indicators including physicochemical parameters [6], faecal indicator bacteria [7,8] and polycyclic aromatic hydrocarbons (PAHs) [9]. The latter pose a serious risk to humans, and are linked with infection or illness from waterborne pathogens during contact-recreation [10]. Guideline values for fresh water recreational water were first presented at the World Health Organization (WHO, 2003),

based on a tolerable impact of <1–5% gastrointestinal disease for voluntary recreational activities, and many countries use *Escherichia coli* (*E. coli*) bacteria as an indicator in fresh water with a 100 cfu/100 mL threshold of risk, based on findings of [11]. The bacterial concentration in rivers is determined by faecal pollution from point and diffuse sources [12]. *E. coli* bacteria replaced faecal coliforms as a more purposeful index of faecal pollution because *E. coli* specifically indicate the presence of faeces from warm-blooded animals. It is considered an indicator of recent faecal pollution due to its higher decay rate than intestinal *Enterococci* in water [12]. In 2012, the Environmental Protection Agency (EPA) updated the water quality criteria recommendations for recreational waters, including both *Escherichia coli* and *Enterococci* as indicators for faecal contamination. The new criteria are designed to protect primary contact recreational uses (e.g., swimming, bathing, surfing, water skiing, water play by children etc.) in which a high degree of bodily contact with water, immersion and ingestion are likely. These recommendations depend on the latest research and science that show a link between gastrointestinal and respiratory illnesses and faecal contamination in recreational waters. *E. coli* guidelines for recreational water quality criteria have been developed by New Zealand (MfE, 2003), EU (Directive 2006/7/EC), the World Health Organization (WHO, 2021), Australia (NHMRC, 2008), the United States Environmental Protection Agency (USEPA, 2012) and Canada (HC, 2012). MfE, 2003 sets the recommended standard of *E. coli* (maximum concentration) in recreational water at 550 cfu/100 mL based upon a statistical threshold value of the 95th percentile, while EU sets 900 cfu/100 mL based upon a 90th percentile evaluation. Additionally, WHO and NHMRC set the standard at 500 cfu/100 mL based upon a statistical threshold value of the 95th percentile. These are draft recommended concentrations to safeguard people from primary contact during recreational use.

The presence of polycyclic aromatic hydrocarbons (PAHs) is also examined within the context of this paper. PAHs are included in the list of priority substances of the Water Framework Directive (WFD) 2000/60/EC, established in the No 2455/2001/EC amendment. PAHs are hazardous organic compounds released into the environment by natural and anthropogenic combustion processes alike. Considerable attention has been paid to PAHs due to their documented carcinogenicity [13] and their persistent nature. Natural sources include volcanic eruptions and forest fires, while incomplete combustion of transportation fuel, industrial waste, agricultural fires and waste incineration constitute examples of anthropogenic sources [9,14].

The purpose of this preliminary study is to investigate the impacts of urbanization on river pollution and the implications for recreational use. The main objective is to assess the physicochemical and microbial quality of the Kifisos river in Attica Greece, which comprises a typical example of an urban river prone to environmental degradation mainly due to its proximity to several pollution sources. We should point out that the Kifisos river is the main urban river of the capital city of Greece (Athens), which nevertheless is not monitored adequately regarding water pollution, and therefore, relevant information is scarce. Thus, this effort is likely the first attempt to present a recent spatiotemporal analysis of the water quality status of the river by using several variables and attempting to identify pollution hotspots along its course. To our knowledge, there is a lack of research on the relationships between physicochemical parameters, nutrients, and *E. coli* concentrations in the Hellenic Water Quality Index (HWQI), and this fact emphasizes the necessity of this study. Descriptive statistics, box plots and correlation matrixes are employed. We analyze the following groups of pollutants: nutrients, pathogens and physicochemical parameters using monthly water surface samples from May 2021 to April 2022 from five sampling sites. Measurements of total polycyclic aromatic hydrocarbons (TPAHs) and total hydrocarbons (THs) were also collected in two of the campaigns in May 2021 and September 2022. Additionally, *E. coli* monitoring results are corelated with the Hellenic Water Quality Index (HWQI) [15–19] according to the demands of the WFD 2000/60 EC to investigate potential relationships. This is a first attempt to investigate correlations between *E. coli* and HWQI in an urban river, which is crucial for future water quality assessments.

The data presented here can be used to provide valuable insights and contribute to the development of comprehensive and inclusive management and restoration plans to tackle the degradation of urban water bodies and their in-stream habitats.

## 2. Materials and Methods

### 2.1. Study Area and Sample Collection

Kifisos river, including various spellings of the name (Kifissos, Cephissus, Kephisos), flows through Athens city and reaches the Saronic Gulf in the western part of Athens metropolitan area (~3.8 million inhabitants). It operates as major drainage channel for a large part of Athens. Its catchment area extends ~420 km$^2$ with the river main route length being approximately 29 km. Most of the year, water discharge is low (~3.2 m$^3$ s$^{-1}$), but during flood events, the water fluxes may reach up to 1400 m$^3$ s$^{-1}$ [20]. According to a previous study [21], certain sections of the Kifisos drainage network, particularly in the central and north-western parts, are vulnerable to flood phenomena. Surface elevation differs between upland and low-land areas, ranging from 1350 m to sea level elevation in its estuary, while the mean elevation of the catchment area is 284.9 m. According to previous studies, approximately 65% of the geological formations are alpine, and the rest are meta-alpine formations [22].

Concerning land use, Kifisos catchment is characterized by multiple anthropogenic pressures, mainly industry, agricultural and residential use. The built-up areas (houses, roads, industries) cover most of the downstream parts of the catchment, and are estimated to exceed 70% while they are still increasing [23]. At two distinct parts along the Kifisos route, industries and manufacturing businesses have been operating on both sides of the river, functioning as unplanned and informal industrial zones. In the upstream areas, forests and shrubland are dominant, but have been significantly impacted by 2007 and 2019 wildfires. This is also where land used for agricultural and livestock use is located. Kifisos has undergone extensive river engineering in the last 13 km of its route, addressing issues of flood control, drainage system management and the road network; Kifisos river is canalized, alternating from fully boxed to open cross-section embankment. As a result, its riparian zone for this stretch was practically eliminated. Moreover, Kifisos Avenue a high-traffic-density road, and the central axis of Athens highway network was constructed over the canalized river. Five sampling locations were selected that are distributed along the drainage catchment from its upstream part, where limited pollution sources exist, to the river mouth (Figure 1).

Monthly water surface collection was conducted in between 10:00 a.m. and 1:00 p.m. from May 2021 to April 2022 at five sampling sites (K Ekv, K 5, K 12, K 17, K MD). Physico-chemical parameters such as dissolved oxygen (DO) (mg/L), electrical conductivity (EC) (µS/cm), temperature (T) (°C), pH and salinity (ppt) were measured on-site by a portable water quality multiparameter instrument (YSI ProDSS). Surface water samples were collected in sterile bottles, preserved by the addition of HgCl$_2$, stored in a cooler box and subsequently transferred to the Hellenic Centre for Marine Research (HCMR) laboratories to further analyze concentrations of NO$_3^-$-N (mg/L), NO$_2^-$-N (mg/L), DIN (mg/L), TN (mg/L) and TP (mg/L). Additionally, the output of the analyses was used to estimate the Hellenic Water Quality Index (HWQI) according to a Nutrient Classification System (NCS) initially developed by [18] and modified to include dissolved oxygen concentrations. Based on the values of five nutrient species along with dissolved oxygen levels, HWQI classifies water samples into five water quality categories: bad, poor, moderate, good and high, according to the requirements of the WFD (2000/60/EC).

For microbiological analysis, water samples were collected separately in 1 L sterile containers for the quantification of *E. coli* according to the International Organization for Standardization (ISO) method/protocol (ISO 9308-1:2014), which specifies the method for the enumeration of *Escherichia coli* (*E. coli*). Two additional water samples were collected in May 2021 and in September 2022 with precleaned glass bottles for TPAHs and THs analysis.

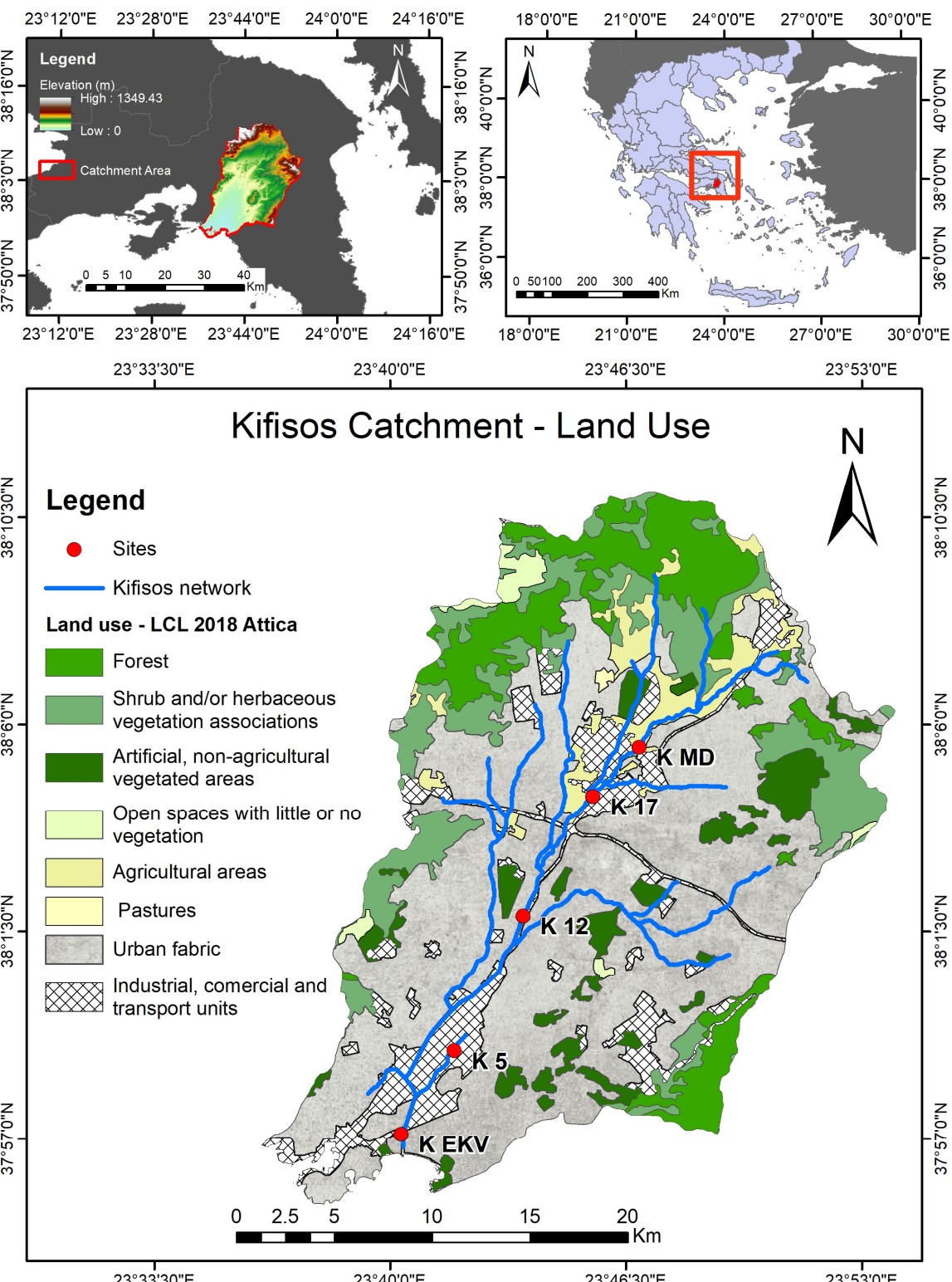

**Figure 1.** Kifisos catchment and location of the five sampling sites (K Ekv, K 5, K 12, K 17, K MD).

### 2.2. Statistical Analysis

First, the Shapiro–Wilk's test was performed to assess for normality of distribution of the river water quality parameters due to the small sample size. Then, the Spearman's non-parametric correlation test was employed to reveal the relationships among the water quality parameters. We analyze the following groups of pollutants: nutrients (nitrate-nitrogen as $NO_3^-$-N (mg/L), nitrite-nitrogen as $NO_2^-$-N (mg/L), ammonium−nitrogen



as $NH_4^+$-N (mg/L), dissolved inorganic nitrogen as DIN (mg/L), total nitrogen as TN (mg/L), orthophosphate as $PO_4$–P (mg/L), total phosphorus as TP (mg/L), pathogens (such as *E. coli*), and physicochemical parameters (dissolved oxygen as DO (mg/L), electrical conductivity as EC (µS/cm), pH, salinity (ppt) and temperature as T (°C) using monthly water surface samples collected from May 2021 to April 2022 from five sampling sites. The data obtained were subjected to statistical analysis using Statistical Package for Social Sciences (SPSS), version 28 software.

Box and whisker plots per sampling site were developed for nutrients, electrical conductivity (EC) and temperature (T) using an open-source application, (BoxPlotR, http://boxplot.tyerslab.com/, accessed on 08 March 2023), to examine the distribution of the measured concentrations. Information about sample sizes was represented by the width of each box, where the widths are proportional to the square roots of the number of observations. Box and whiskers plots present the min-max and median in addition to the first and third quartiles of each variable and site for the entire sampling period.

Due to the large variability in the *E. coli* concentrations, the visualization of this pathogen was made via beanplots to provide a clear view of the central tendency and variability of the data. In the beanplots, the individual observations are presented as small white lines. Results were compared with EU recommendations (Directive 2006/7/EC; Annex 1 for inland waters). Comparisons were made based on the Hazen method according to the addendum of the World Health Organization (2009).

Quality classes were assigned to the sampling sites according to the Hellenic Water Quality Index (HWQI). Finally, Spearman's correlation was applied between the study variables to check for covariances. Correlations were used to identify monotonic relationships between two quantitative parameters simultaneously using the 1-tailed Spearman's rank correlation. The criterion used to check the reliability was the correlation coefficient (ρ).

## 3. Results

### 3.1. Correlations between the Examined Water Quality Parameters along Kifisos River

Due to of the small size of the sample data, we used the Shapiro—Wilk test to test for normality of distributions of the following water quality parameters (Table 1). The $p$ values were considered statistically significant at a 95% confidence interval. According to the results of the Shapiro–Wilk test for normal distribution, most of the examined water quality parameters are not normally distributed.

**Table 1.** Test of normality (Shapiro–Wilk).

| No | Parameter (Unit) | Statistic | $p$ Value * |
|----|------------------|-----------|-------------|
| 1 | pH | 0.972 | 0.182 |
| 2 | DO (mg/L) | 0.971 | 0.160 |
| 3 | DIN (mg/L) | 0.961 | 0.054 |
| 4 | **Salinity (ppt)** | **0.524** | **<0.001** |
| 5 | **TN (mg/L)** | **0.953** | **0.027** |
| 6 | **TP (mg/L)** | **0.917** | **<0.001** |
| 7 | **T (°C)** | **0.952** | **0.019** |
| 8 | **EC (µS/cm)** | **0.539** | **<0.001** |
| 9 | **$NO_3^-$-N (mg/L)** | **0.944** | **0.006** |
| 10 | **$NO_2^-$-N (mg/L)** | **0.508** | **<0.001** |
| 11 | **$NH_4^+$-N (mg/L)** | **0.551** | **<0.001** |
| 12 | **$PO_4$-P (mg/L)** | **0.460** | **<0.001** |
| 13 | ***E. coli* (cfu/100 mL)** | **0.604** | **<0.001** |

* $p$ values in bold were considered statistically significant at a 95% confidence interval.

As the majority of the data did not fit a normal distribution, Spearman's correlation analysis was then applied to check for covariances. Spearman's correlation analysis showed that temperature has significant ($p < 0.05$) correlations with nutrients, particularly with phosphorus (Table 2). Regression analysis for orthophosphates and total phosphorous as

function of temperature yielded values ($R^2$) 0.347 and 0.346, respectively. Figure 2 also indicates a strong seasonal trend among temperature and high nutrient values, especially in the summer season (upward-sloping power curve; June–July–August).

**Table 2.** Spearman's correlation among the examined parameters (bold numbers are statistically significant with $p < 0.05$) *.

| | pH | DO | DIN | Salinity | TN | TP | T | EC | $NO_3^--N$ | $NO_2^--N$ | $NH_4^+-N$ | PO4-P | *E coli* | HWQI |
|---|---|---|---|---|---|---|---|---|---|---|---|---|---|---|
| **pH** | 1.00 | | | | | | | | | | | | | |
| **DO** | **0.61** | 1.00 | | | | | | | | | | | | |
| **DIN** | −0.06 | **−0.31** | 1.00 | | | | | | | | | | | |
| **Salinity** | **−0.42** | **−0.31** | 0.07 | 1.00 | | | | | | | | | | |
| **TN** | −0.04 | **−0.29** | −0.01 | 0.03 | 1.00 | | | | | | | | | |
| **TP** | −0.07 | **−0.31** | **0.49** | −0.12 | **0.49** | −0.23 | | | | | | | | |
| **T** | **−0.25** | **−0.49** | **0.27** | 0.09 | **0.28** | **0.54** | 1.00 | | | | | | | |
| **EC** | −0.39 | **−0.30** | 0.06 | **0.96** | 0.03 | −0.07 | 0.14 | 1.00 | | | | | | |
| **$NO_3^--N$** | 0.03 | 0.01 | **0.56** | **0.38** | **0.54** | 0.11 | 0.04 | **0.40** | 1.00 | | | | | |
| **$NO_2^--N$** | −0.17 | **−0.43** | **0.59** | 0.10 | **0.60** | 0.33 | **0.27** | 0.10 | 0.06 | 1.00 | | | | |
| **$NH_4^+-N$** | −0.12 | **−0.32** | **0.40** | −0.10 | **0.42** | 0.20 | 0.10 | −0.11 | −0.23 | **0.81** | 1.00 | | | |
| **PO4-P** | −0.08 | **−0.32** | **0.46** | −0.10 | **0.46** | **0.99** | **0.53** | −0.05 | 0.11 | 0.30 | 0.18 | 1.00 | | |
| ***E coli*** | 0.00 | **−0.35** | 0.16 | −0.18 | 0.19 | −0.03 | 0.16 | −0.19 | −0.17 | **0.45** | **0.48** | −0.06 | 1.00 | |
| **HWQI** | 0.05 | **0.57** | **−0.52** | −0.01 | **−0.54** | **−0.57** | **−0.44** | −0.07 | −0.09 | **−0.76** | **−0.69** | **−0.55** | **−0.29** | 1.00 |

* The data supporting the findings of this study are described in Supplementary File Table S1 (Supplementary Materials).

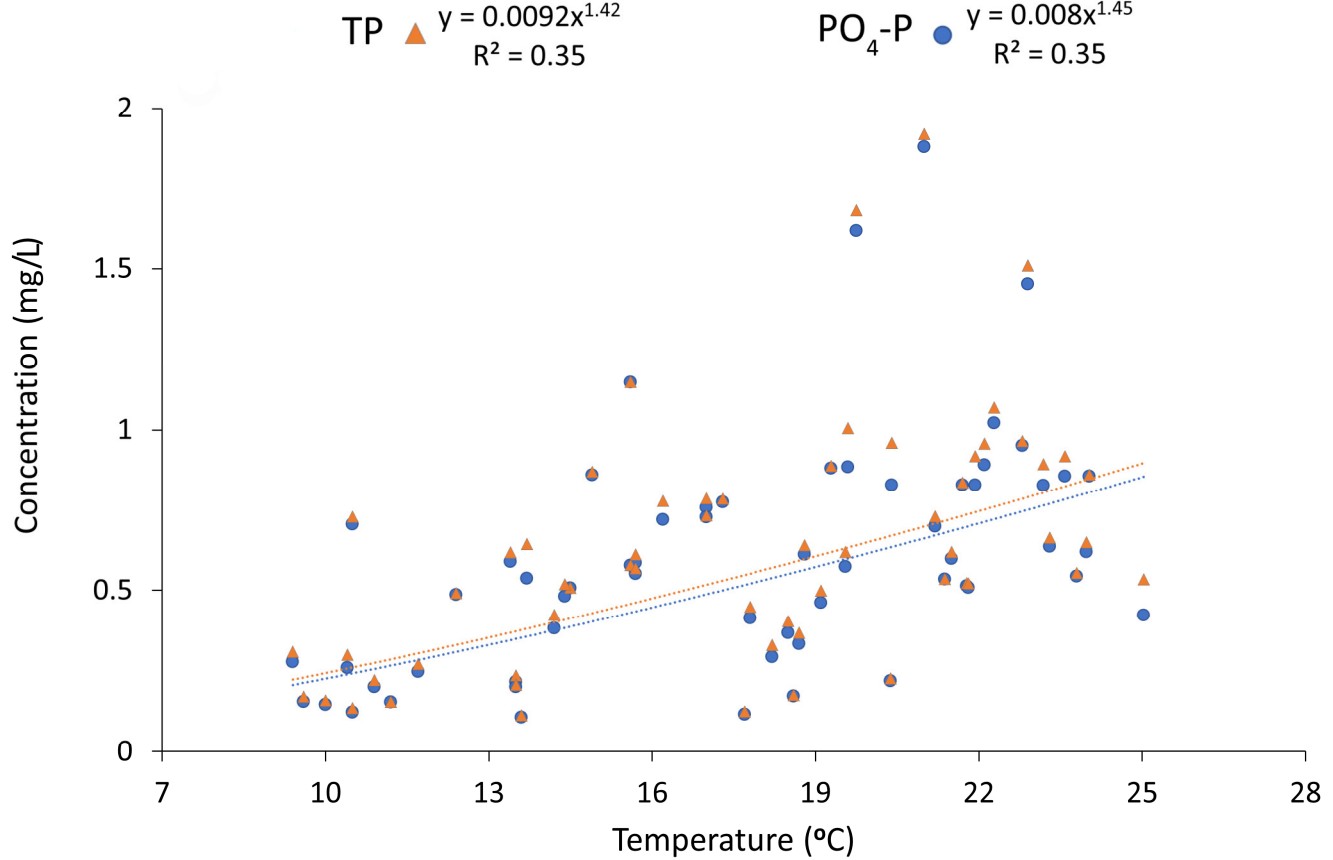

**Figure 2.** Variation of temperature with orthophosphates ($PO_4$-P) and total phosphorus (TP) in Kifisos river.

HWQI and temperature are also strongly correlated, confirming that during warm periods, the physicochemical quality gets worse. The DO concentration is also strongly correlated with all nutrients, HWQI and *E. coli*, which implies that when these substances' levels increase, dissolved oxygen is decreased. This is expected since organic pollution, which also consumes dissolved oxygen due to intense microbial activity, is the main factor affecting these parameters. HWQI and *E. coli* are more strongly correlated with $NO_2^--N$ and $NH_4^+-N$ than with the rest of nutrients, likely implying the frequent presence of

human sewage in the river. Nevertheless, there are other sources of nutrients as well since the correlation between *E. coli*, $NO_3^-$-N and HWQI is not very strong. The very high correlations between dissolved inorganic nitrogen (DIN) and total nitrogen (TN) as well as between total phosphorus (TP) and $PO_4$-P indicates anthropogenic pollution as the dominant source of nutrients in the river (Table 2). In Table 2 green–yellow–red color scale was applied to visually represent data values based on their relative magnitude compared to the rest of the values. The lowest values were colored with red shades, the middle values with yellow shades, and the highest values with green shades.

### 3.2. Nutrients' Statistics per Site

Nitrogen-nitrate in almost all samples was very high, with values ranging from close to 0 up to 20 mg/L (Figure 3). More than 90% of the total samples fell in the category of "bad quality" for $NO_3^-$-N, and the sites having more than 50% of their samples with very high concentrations (more than 7 mg/L) are K 12 (approx. in the middle of the catchment) and K Ekv (close to the river mouth). K MD, which is in the upstream area, had the lowest $NO_3^-$-N values, but those values were still high enough to be classified as "bad quality." Total nitrogen (TN) values were very similar to $NO_3^-$-N values in most cases, which means that most of the nitrogen-related pollution is inorganic and in the form of nitrates, while a significant difference is observed at the K 5 site since TN values are generally much higher than $NO_3^-$-N concentrations (Figure 3). This is due to the fact that $NH4^+$-N values are also very high at the specific site, which could be related to domestic or industrial sewage in close vicinity to the sampling area.

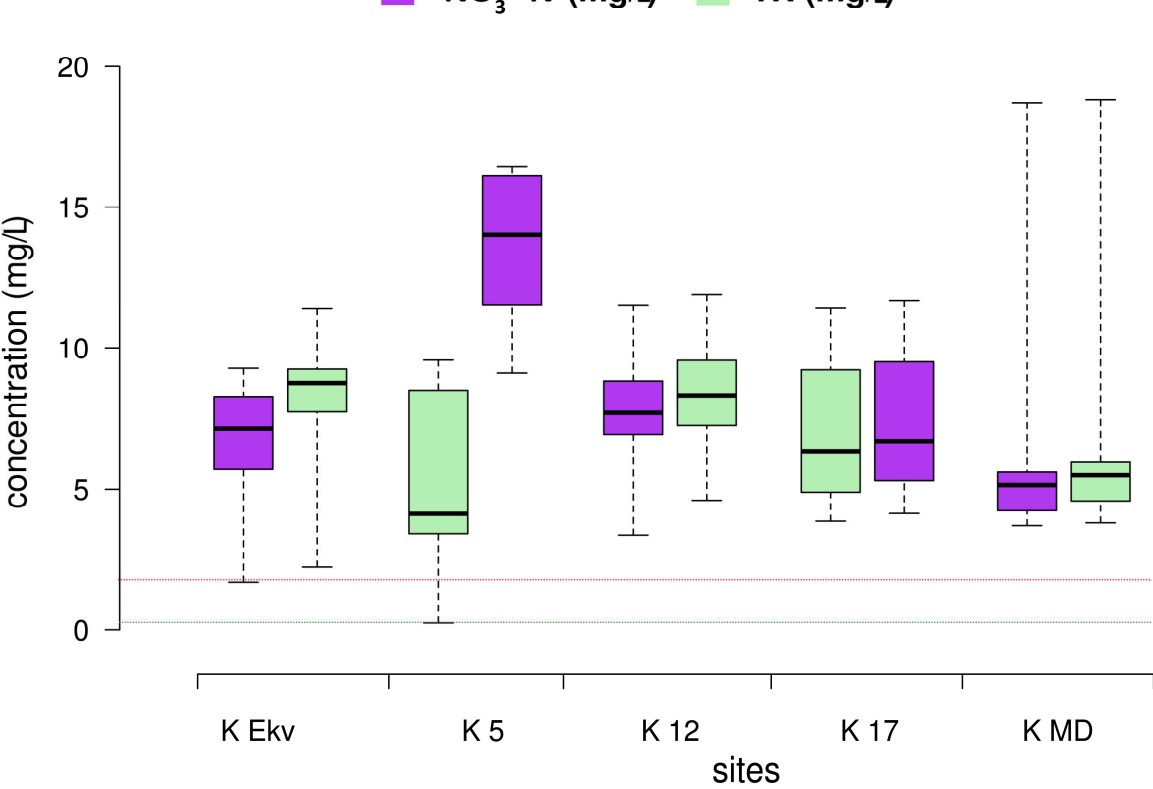

**Figure 3.** Distribution of $NO_3^-$-N and TN values at the sampling sites from upstream (K MD) to the river mouth (K Ekv). The dotted lines indicate good (green) and bad (red) thresholds for $NO_3^-$-N according to the WFD provisions.

Similarly, for phosphorus, most of the samples indicated high concentrations and therefore poor and bad quality. Only the river mouth area (K Ekv) presented a slightly better picture, with more than 50% of the samples to fluctuate between the moderate and

poor classifications (Figure 4). This is likely due to the effect of the sea, which generally has significantly lower P concentrations, and therefore, the mixing processes that occur at K Ekv improve slightly the quality of the site. The sites that present the highest $PO_4$-P and TP concentrations are K 17 (close to the industrial area of Metamorphosis) and K 5 (in another industrial area close to Athens city center) that also had very high $NH_4^+$-N levels. Nevertheless, even in the upstream part of the catchment (K MD), more than 75% of the samples were classified as of bad quality.

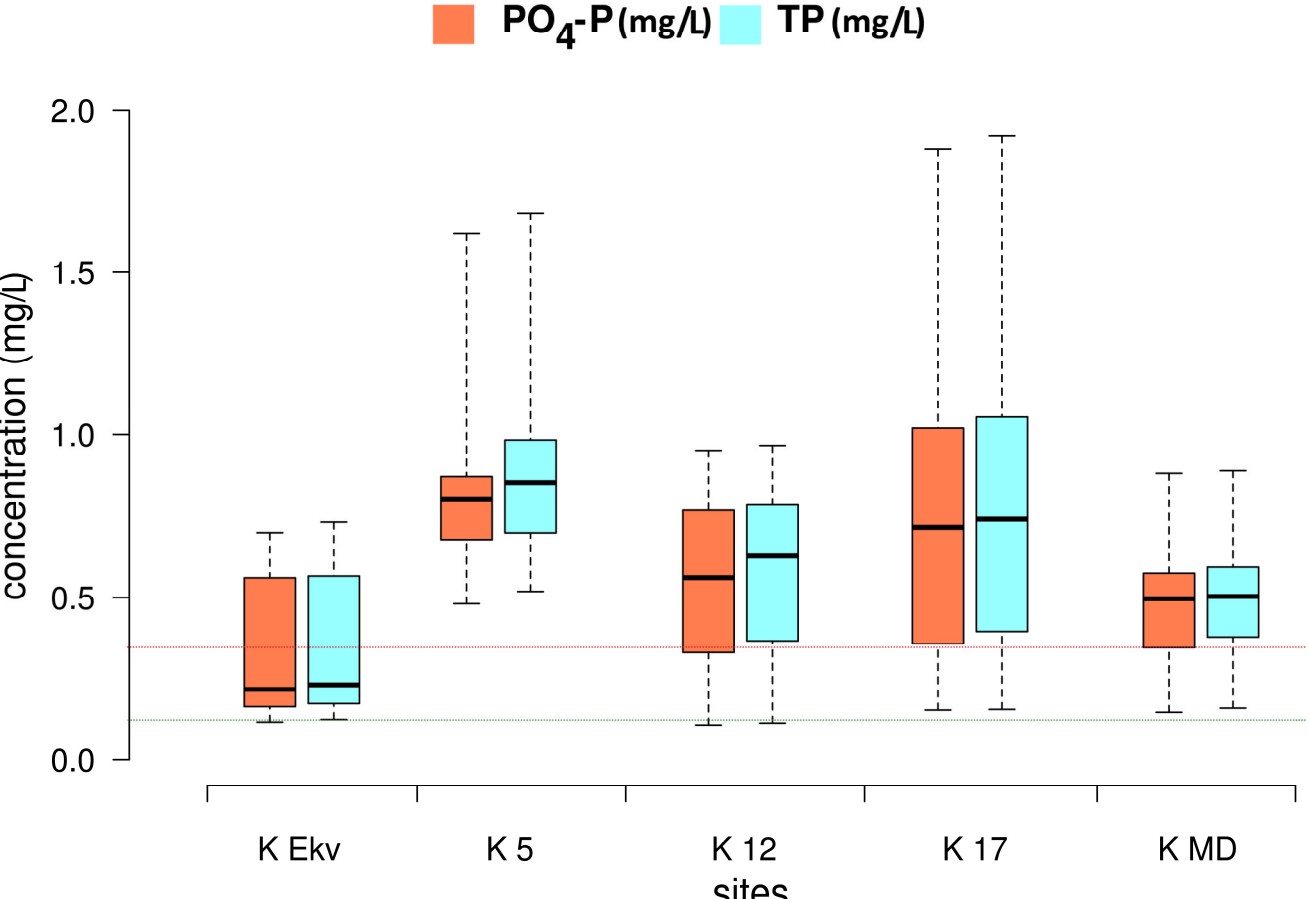

**Figure 4.** Distribution of $PO_4$-P and TP values in the sites from upstream (K MD) to the river mouth (K Ekv). The dotted lines indicate good (green) and bad (red) thresholds for $PO_4$-P according to the WFD provisions.

According to the HWQI, the sampling sites K Ekv, K 12 and K 17 could be characterized as of bad or poor physicochemical status in the majority of the measurements. Approximately 50% of the samples in K MD (upstream) and 25% of the samples in K Ekv and K 12 indicated moderate physicochemical quality (Figure 5). The worst results in terms of WFD quality class were recorded in K 5, with more than 50% of the samples classified as of bad quality and all the rest as of poor quality. K 17 follows, with almost 80% of the samples classified as of poor physicochemical quality.

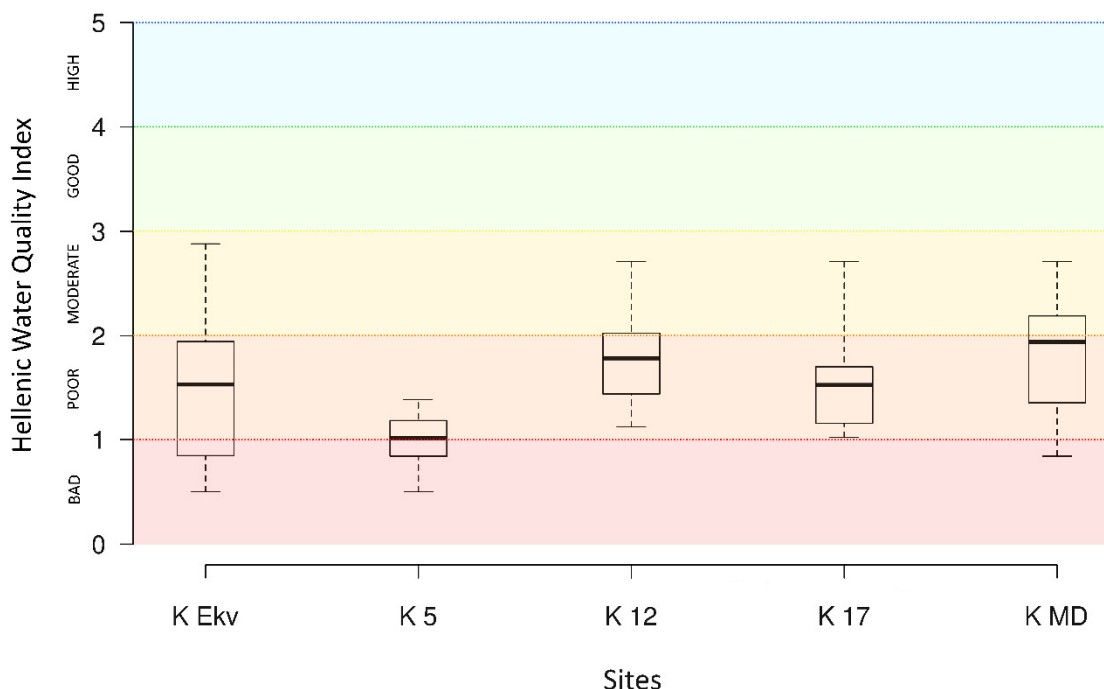

**Figure 5.** Box plots per sampling site for the visual representation of the HWQI (HWQI estimated with the use of six water parameters: five nutrient species and DO; colors that indicate the relative water quality conditions: red = bad, orange = poor, yellow = moderate, green = good and blue = high).

*3.3. Physicochemical Fluctuations Analysis per Site*

Figure 6 represents electrical conductivity (EC) (left axis) and temperature (T) (right axis) observations per sampling site. It was observed that the EC concentrations ranged from 286.3 μS/cm to 8146 μS/cm in water obtained from K Ekv. The wide variation at this sampling site, which is considerably different from the rest, is mainly due to seawater influence. The highest EC values in most all other cases were close to 1000 μS/cm. Such a value is within the expected range for fresh water in an urban river. K 17 and K MD had proximity in the EC minimum values (K 17: 513 μS/cm and K MD: 519 μS/cm), median (K 17: 1012 μS/cm and K MD: 1043.5 μS/cm) and maximum values (K 17: 1117 μS/cm and K MD: 1160 μS/cm). Water temperatures ranged from 9.4 to 25 (°C), which would pose no adverse effects on living organisms. K 12 and K 17 have proximity in their values, while K MD maximum, median and minimum values are lower compared to the other four sites.

*3.4. Variations of E. coli Concentrations per Site*

Sampling sites of the Kifisos river had a large variation in *E. coli* concentrations during the examined period (Figure 7). A comparative assessment was made using the Hazen method, as recommended by WHO 2009, and results are presented in Table 3. The maximum admissible number as set by the EU Directive 2006/7/EC indicative of good quality conditions was exceeded on all occasions, with the highest value found being 117,200 (cfu/100 mL) at the K5 site.

**Table 3.** Hazen method results for *E. coli* values (cfu/100 mL) for each site of Kifisos river.

| Site | K EKV | K 5 | K 12 | K 17 | K MD |
|---|---|---|---|---|---|
| **Percentile** | 95 | 95 | 95 | 95 | 95 |
| **Data minimum** | 700 | 1640 | 500 | 200 | 100 |
| **Data maximum** | 34,400 | 120,000 | 100,000 | 98,000 | 2700 |
| **Hazen result** | 34,060 | 117,200 | 95,400 | 93,100 | 2616 |

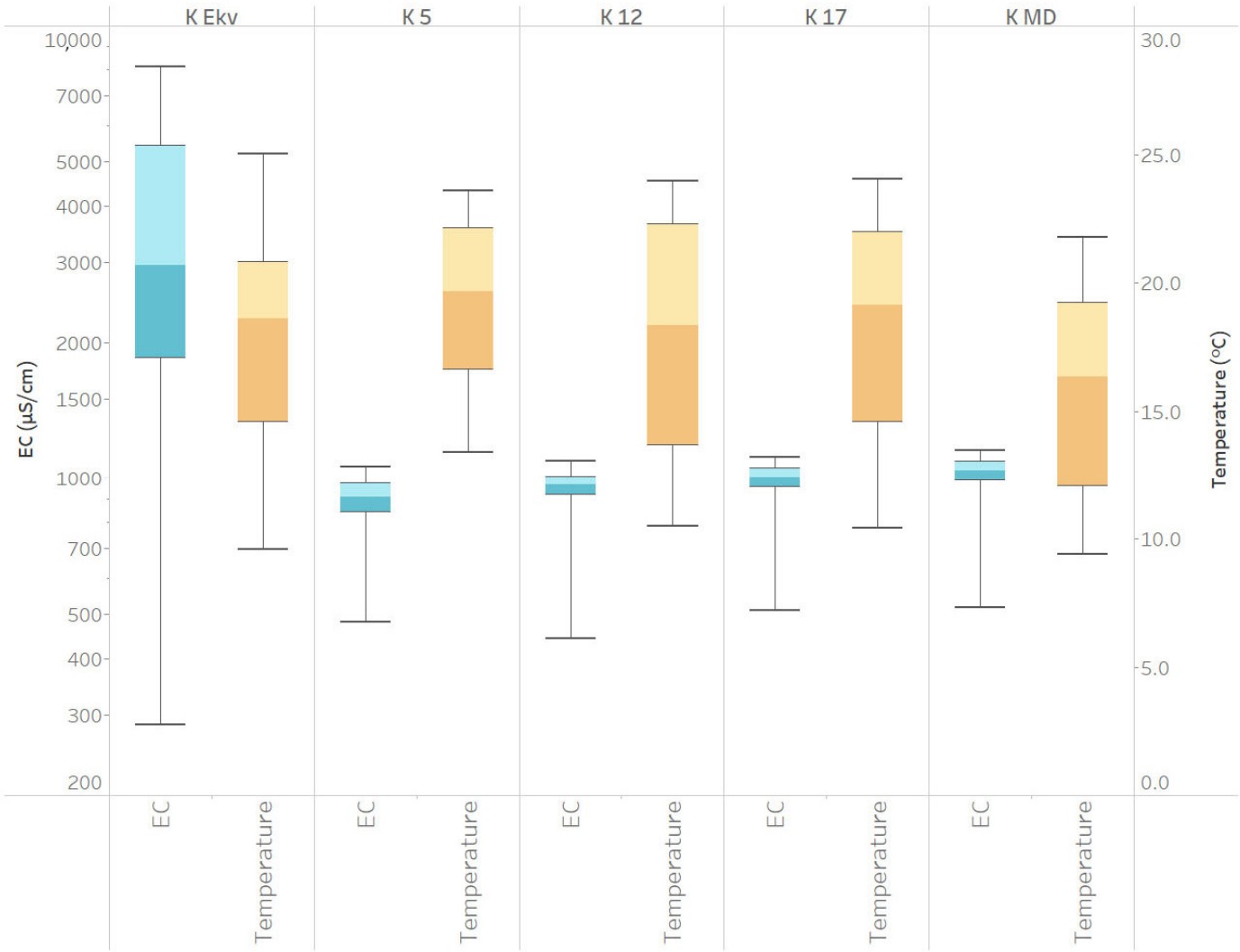

**Figure 6.** Box plots per sampling site for electrical conductivity (blue grading plots) and temperature (yellow grading plots). Light colors represent values above the median and dark colors represent values below the median; box limits indicate the 25th and 75th percentiles; whiskers extend to minimum and maximum values.

Extremely high values were also found at the K 12 and K 17 sites (100,000 cfu/100 mL), while the lowest median value of *E. coli* was observed at K MD (Figure 7). These results are in accordance with the nutrient findings mentioned above since there is a tendency for pollution levels to increase from the upstream to the downstream part of the river, with the exception of K 5, which is a small branch of the river that is inside an industrial area and is heavily polluted.

Moreover, the Pearson correlation coefficient (r) per site was calculated (Table 4) to examine the patterns between *E. coli* and the physicochemical parameters (T, EC, pH and DO). A strong negative correlation was estimated between EC and *E. coli* for two (K 12 and K MD) out of five sites. For these two sites, EC and *E. coli* concentration linear regression models were developed. These models resulted in a coefficient of determination R-squared ($R^2$) of 0.63 for the K MD site and an $R^2$ of 0.15 for K 12.

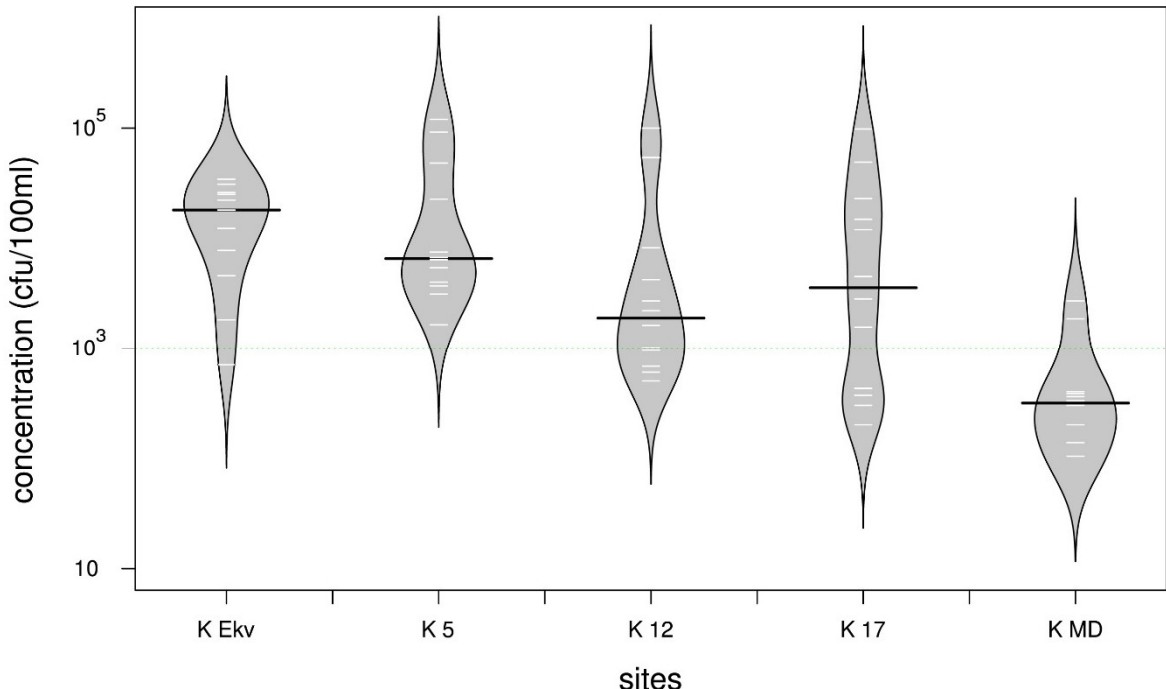

**Figure 7.** Black lines show the medians; white lines represent individual data points, while the gray area shows the distributions. Green line indicates the good quality threshold according to the EU Directive 2006/7/EC.

**Table 4.** Pearson correlation coefficient (r) per site between *E. coli* and the physicochemical parameters (bold values are statistically significant with $p < 0.01$).

| Site/r | T (°C) | EC (μS/cm) | pH | DO (mg/L) |
|---|---|---|---|---|
| K Ekv | 0.1 | 0.1 | 0.2 | 0.0 |
| K 5 | −0.2 | −0.2 | 0.1 | −0.2 |
| K 12 | −0.3 | **−0.8** | 0.0 | −0.1 |
| K 17 | −0.3 | −0.1 | 0.1 | 0.2 |
| K MD | −0.1 | **−0.8** | −0.3 | 0.0 |

TPAHs concentrations also indicated the existence of heavy pollution, and ranged from 17.2 (K MD) to 4541 ng/L (K 5), with a mean value of 764.3 ng/L. Additionally, TPAHs concentrations ranged from 8.1 (K MD) to 1050 μg/L (K 5), with a mean value of 313.8 μg/L (Table 5).

**Table 5.** Total polycyclic aromatic hydrocarbons (TPAHs) and total hydrocarbons (THs) concentrations.

| Site | TPAHs (ng/L) | | THs (μg/L) | |
|---|---|---|---|---|
| | September 2021 | February 2022 | September 2021 | February 2022 |
| **K Ekv** | 1272.0 | 112.7 | 382.8 | 20.4 |
| **K 5** | 4541.6 | 402.8 | 1050 | 84.2 |
| **K 12** | 147.5 | 168.9 | 91.2 | 54.1 |
| **K 17** | 106.8 | 728.0 | 924 | 468.0 |
| **K MD** | 145.7 | 17.2 | 55.2 | 8.1 |

## 4. Discussion

As demonstrated by other researchers [3,8], the quality of recreational water must be of prescribed standards for both physicochemical and microbial quality. According to similar

studies for urban rivers, Kifisos is not bathing water. However, it is used for recreational and commercial activities, where users can get in direct contact with the water via several activities (rowing, boating, kayaking, fishing and activities on the banks), making the Water Framework (2000/60/EC) as well as the bathing water (2006/7/EC) directives appropriate reference for risk assessment [7,24]. Additionally, the physicochemical water quality status of all five study locations during the sampling period was always below good threshold according to the WFD standards. Similar water quality conditions were estimated in previous studies for the Kifisos river [15].

Analysis with Spearman's correlation demonstrated that the physicochemical water quality index (HWQI) was significantly correlated with $NO_2^--N$ and $NH4^+-N$ and to a lesser degree PO4-P, which implies the substantial presence of human sewage in the river system. This is attributed to the very high levels of *E. coli,* for which the maximum admissible *E. coli* concentration, as set by the EU directives, was exceeded on all occasions. EC can indirectly influence the presence and behavior of *E. coli* since it is often associated with the concentration of dissolved ions in water, including nutrients like phosphates and nitrates, which may support the growth and survival of *E. coli* bacteria. These contaminants could originate from various sources, such as industrial discharges or agricultural runoff, and can affect the viability of *E. coli* populations. However, EC is not a direct measure of *E. coli* presence or concentration. A coefficient of determination R-squared ($R^2$) of 0.63 for the K MD site indicated that the two variables are likely related to each other. However, more data are necessary for decisive validation of the relationships illuminated in this study.

Pollution of the Kifisos river with human sewage likely occurs when the sewer system gets overloaded during heavy rainfall events, during illegal disposal of industrial and domestic waste, livestock farming effluents and other illegal point pollution sources. Other studies [25,26] found similar causes that led to water quality deterioration in urban rivers. Citizen engagement is important to monitor and improve river water quality effectively and achieve the goal of a progressive reduction in discharges from the point and non-point pollution sources, coupled with knowledge on factors that are related to urban riverscapes and recreational uses [27].

We conducted this preliminary study examining the Kifisos river to analyze its water quality status, including physicochemical parameters, nutrients, *E. coli* and polycyclic aromatic hydrocarbons concentrations. The results indicated the significant water quality degradation of the river, particularly at downstream stations, between the stations K 17 and K Ekv, probably triggered by urban runoff, illegal sewage and industrial waste disposal. K 17 and K 5 are the stations with the worse quality status, which was expected due to their close proximity to industrial areas. Even the upstream station (K MD) illustrated a poor or bad physicochemical water quality status in more than 50% of the sampling efforts. An existing study [28] for the Kifisos river revealed similar influence by anthropogenic activities regarding major elements (nitrate and phosphate concentrations). Another study [29] revealed that the Kifisos river is highly polluted and ranked the river's quality as very poor regarding species diversity. In the current study, the average *E. coli* concentration was significantly high at all stations, characterizing the water as bad quality, while the total polycyclic aromatic hydrocarbons (TPAHs) results indicated the existence of significant pollution. More specifically, the TPAHs and TH concentrations detected along the river (particularly in K 5 site) indicated the presence of pollutants originating from petroleum sources. Our results regarding the levels of the TPAHs and THs are similar to other studies in urban areas [16,30]. High TPAHs and THs values should most likely be attributed to road runoff from heavy traffic highways adjacent to the river. Point source pollution from industries located along the route of the river should also be examined. Moreover, other manmade chemicals, such as heavy metals and pesticides, can also impact water quality and aquatic ecosystems. Therefore, monitoring should be done in conjunction with other water quality assessments to provide a comprehensive evaluation of river health. All of the above indicate the necessity of continuous environmental monitoring at certain points of

the river and legal investigations to identify the specific polluters in order to undertake the necessary restoration actions.

In the future, a multi-pollutant modelling approach would help to better understand and manage water quality issues, since monthly sampling is unlikely to offer the necessary information for classifying the water quality status accurately [31]. These issues can be addressed by field-deployable water quality analyzers designed to continuously measure concentrations of pollutants in freshwater [32]. Switching to sustainable uses of freshwater resources is critical for minimizing land degradation and biodiversity loss [33].

From a water quality management perspective, it is evident that continuous environmental monitoring at certain points of the rivers is essential to reduce uncertainty in the results. High frequency samplings with the adoption of modern sensors can provide high frequency data [34] and substantially larger datasets to reduce uncertainty of pollution load estimates, particularly in urban rivers with multiple stressors, such as the Kifisos river [35].

## 5. Conclusions

This study is a scientific attempt to understand the dominant pollution pressures and their spatiotemporal patterns in the Kifisos river in the context of the EU Water Framework Directive. Bacteriological quality of the water in the Kifisos river was unacceptable, as the study demonstrated the widespread occurrence of *E. coli* and low-quality physiochemical conditions. There is no clear cut-off value at which health effects are excluded. However, the increased microbial numbers can cause recreational water quality deterioration, and it is a major problem for urban rivers such as the Kifisos river. In order to avoid further degradation and continuous decline of the ecological status of the Kifisos river, several protection and restoration measures should be undertaken within the framework of the existing EU and national policies. Additionally, we recommend legal investigations to identify polluters and undertake the necessary legal and environmental restoration actions. Surface runoff should be also included in future studies to investigate the accuracy and description of pollutant transfer processes during rainfall events and their effect on the total pollutant load of urban rivers.

**Supplementary Materials:** The following supporting information can be downloaded at: https://www.mdpi.com/article/10.3390/app13116698/s1, Table S1: Physicochemical data of Temperature (T), Electrical Conductivity (EC), Salinity, pH, Dissolved Oxygen (DO) and nitrate-nitrogen as $NO_3^-$-N, nitrite-nitrogen as $NO_2^-$-N, Ammonium-Nitrogen as $NH_4^+$-N, Dissolved Inorganic Nitrogen as DIN, Total Nitrogen as TN, Orthophosphate as $PO_4$-P, Total Phosphorus as TP, *Escherichia Coli* as *E. coli* and Hellenic Water Quality Index as HWQI.

**Author Contributions:** Conceptualization, C.P. and E.D.; Investigation, S.L.; Resources, E.D.; Data curation, C.P. and E.D.; Writing—original draft, C.P.; Writing—review & editing, S.L. and E.D.; Visualization, C.P. All authors have read and agreed to the published version of the manuscript.

**Funding:** This research received no external funding.

**Institutional Review Board Statement:** Not applicable.

**Data Availability Statement:** Data is contained within the Supplementary Materials Table S1.

**Conflicts of Interest:** The authors declare no conflict of interest.

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
