# Peer review of "Preliminary Analysis of the Water Quality Status in an Urban Mediterranean River"

_applsci, doi:10.3390/app13116698_

Round 1
Reviewer 1 Report
This manuscript reports on the water quality of the Kifisos river in Greece from its headwaters to its inlet at five stations during different months of the year. Although the data provide some feedback, the overall novelty is lacking and many of the results and impacts are predictable. If the author still wants to publish this manuscript, I have some suggestions:
1. Please provide all raw data of different parameters for each month in the form of a supplementary file.
2. Please double check the appropriateness of the cited paper, for example, lines 75 to 77, I have checked the cited paper and there is no mention of volcanic eruptions and agricultural fires as the main natural causes, please read the cited paper again carefully.
3. Please improve the clarity and quality of the figures in the manuscript, such as figure 2 and figure 5.
4. Some microbial generic names are not in italics, such as line 58, and some punctuation mark is abused. Please check the manuscript again.
5. In the discussion section, it is necessary to compare the research results of this manuscript with those of published articles and explain the reasons for similarities or differences. The conclusion section should summarize the specific results and unique findings of this manuscript, rather than describing some common sense such as "Urban diffuse pollution from cities and transport, carries pollutants from roads and urban surfaces into storm drains and out into watercourses at many locations along the river."
Author Response
We appreciate the time and effort that the reviewer has dedicated to providing his/her valuable feedback giving us the opportunity to improve our manuscript.
Reviewer #1
This manuscript reports on the water quality of the Kifisos river in Greece from its headwaters to its inlet at five stations during different months of the year. Although the data provide some feedback, the overall novelty is lacking and many of the results and impacts are predictable. If the author still wants to publish this manuscript, I have some suggestions:
- Please provide all raw data of different parameters for each month in the form of a supplementary file.
We created a new supplementary file (Table S1) including the data supporting the findings of this study. Line 221 in revised manuscript (word version, track changes on).
- Please double check the appropriateness of the cited paper, for example, lines 75 to 77, I have checked the cited paper and there is no mention of volcanic eruptions and agricultural fires as the main natural causes, please read the cited paper again carefully.
We double checked the cited paper of Manoli & Samara, (1999). More specifically it is mentioned in the introduction section that: “Volcanic eruptions and forest and prairie fires are among the major natural sources of PAHs in the atmosphere.”
- Please improve the clarity and quality of the figures in the manuscript, such as figure 2 and figure 5.
We improved the clarity and quality of the specified figures 2 and 5.
- Some microbial generic names are not in italics, such as line 58, and some punctuation mark is abused. Please check the manuscript again.
Microbial generic names have been replaced with italics throughout the revised manuscript. We also fixed the punctuation marks.
- In the discussion section, it is necessary to compare the research results of this manuscript with those of published articles and explain the reasons for similarities or differences. The conclusion section should summarize the specific results and unique findings of this manuscript, rather than describing some common sense such as "Urban diffuse pollution from cities and transport, carries pollutants from roads and urban surfaces into storm drains and out into watercourses at many locations along the river."
The reviewer is right and, thus, we moved the respective sentence to introduction. Lines 34-36 in revised manuscript (word version, track changes on).
Unfortunately, there are no many studies conducted in the past in Kifisos river and therefore the comparison in the discussion section could only be done with limited relevant research efforts from the international literature (eg. Argyraki Koutsomitrou et al., 2014, Argyraki et al. 2013, Adeniji, Okoh, et al., 2019; Markogianni, Anastasopoulou, et al., 2018, Panagopoulos, Alexakis, et al., 2022).
Reviewer 2 Report
The author have studied the quality of a particular river. My question is, can the author substantiate the necessity of this study? Because, whenever a pollutant, industrial or natural, it is obvious that the water gets polluted and not advised to use for any activity. We just need to find out is there any source that is disposing its waste into the river and we can conclude straight away that it is polluted.
Organic pollutants are generally present in all river bodies, fecal matters of animals and birds cannot be avoided. This is true for all water bodies around the world and it is important for the nitrogen cycle of the water body. I believe that this more important than human recreation and this should not be interfered with.
Author Response
- Responses provided in blue text –
We appreciate the time and effort that the reviewer has dedicated to providing his/her valuable feedback giving us the opportunity to improve our manuscript.
The author have studied the quality of a particular river. My question is, can the author substantiate the necessity of this study? Because, whenever a pollutant, industrial or natural, it is obvious that the water gets polluted and not advised to use for any activity. We just need to find out is there any source that is disposing its waste into the river and we can conclude straight away that it is polluted.
Thank you for your comments the following text has been inserted in the manuscript: “We should point out that Kifisos river is the main urban river of the capital city of Greece (Athens) which nevertheless is not monitored adequately regarding water pollution and therefore the relevant information is scarce. Thus, this effort is probably the first one attempting to present a recent spatiotemporal analysis for the water quality status of the river, by using several variables and attempting to identify pollution hot-spots along its course. To our knowledge, there is a lack of research on the relations between physicochemical parameters, nutrients, and E. coli concentrations with the Hellenic Water Quality Index (HWQI) and this fact emphasizes the necessity of this study.” Lines 84-91 in revised manuscript (word version, track changes on).
Kifisos river comprise a typical example of an urban river prone to environmental degradation mainly due to its proximity to several pollution sources.
In the discussion section we describe pollution sources of the Kifisos river. Lines 364 -368, 371-373 in revised manuscript (word version, track changes on).
Organic pollutants are generally present in all river bodies, fecal matters of animals and birds cannot be avoided. This is true for all water bodies around the world and it is important for the nitrogen cycle of the water body. I believe that this more important than human recreation and this should not be interfered with.
Thank you for your comment. Given the very high concentrations of nutrients and E. coli, the anthropogenic impacts in the water pollution status seems to be significant in the specific case (on top of the existing animals wastes which cannot be the primary source due to the urban character of the river). In the conclusions section: “we recommend legal investigations to identify polluters and undertake the necessary legal and environmental restoration actions.” Lines 420-424.
Reviewer 3 Report
This study has assessed the physicochemical and microbial water quality parameters of an urban Mediterranean river. Overall, this study is important from the environmental implication perspectives. The results were well explained and discussed with support from the statistical analysis. However, the reviewer has provided several comments for consideration of revision to enhance the overall quality of the manuscript and readers understanding of the presented findings.
Comments:
Abstract:
Provide some quantitative data in the abstract.
At the end of the abstract, include important implication of this work.
Line 16: Change “E. Coli” to “E. coli” with italicize the text.
Keywords: Replace “water; physicochemical” by giving more specific keywords.
Italicize “E. coli” and “Enterococci” throughout the manuscript.
Line 63 – 66: Also give quantitative data, i.e., the recommended standard (maximum concentration) of E. coli in recreational water by various countries/international water quality regulatory organizations.
In introduction, authors need to highlight the novelty and importance of this study with respect to the existing findings in the scientific literature.
Line 128: “five sampling sites (K Ekv, K 5, K 12, K 17, K MD).” Could you explain on which basis these five water sampling sites were selected. Was the sampling done on the surface water or certain deep from the surface.
Line 132: “HgCl2,” Check the typo error here. Clearly mention the role of HgCl2, and whether addition of HgCl2 interferes the measurement of the water quality parameters by analytical methods.
Line 259 – 260: “which is within the expected range for freshwater” Show the values for the concentration range here.
Line 308 – 312: Strengthen the discussion on the correlation between physicochemical water quality parameters and microbial (E. coli) water quality, i.e., which of the physicochemical water quality parameters are strongly correlated with E. coli level in each sampling site.
Table 1: The heading (a/a) for the first column is not clear. Change the heading of the second column to “Parameter (Unit)”. Also, in the footnote, mention about the meaning of the bolded texts in the table.
Minor typographical editing is needed throughput the manuscript.
Author Response
- Responses provided in blue text –
We appreciate the time and effort that the reviewer has dedicated to providing his/her valuable feedback giving us the opportunity to improve our manuscript.
Abstract:
Provide some quantitative data in the abstract.
We added quantitative information about E. coli. Lines 15-16, in revised manuscript (word version, track changes on).
At the end of the abstract, include important implication of this work.
We followed reviewer’s recommendations. Lines 16-19, in revised manuscript (word version, track changes on).
Line 16: Change “E. Coli” to “E. coli” with italicize the text.
Done.
Keywords: Replace “water; physicochemical” by giving more specific keywords.
New Keywords: water quality; statistical analysis.
Italicize “E. coli” and “Enterococci” throughout the manuscript.
Done.
Line 63 – 66: Also give quantitative data, i.e., the recommended standard (maximum concentration) of E. coli in recreational water by various countries/international water quality regulatory organizations.
We added additional information as the R 3 indicated. “MfE, 2003 sets recommended standard of E. coli (maximum concentration) in recreational water by 550 cfu/100 ml based upon a statistical threshold value of the 95th percentile, while EU sets 900th cfu/100 ml based upon a 90th percentile evaluation. Additionally, WHO and NHMRC set 500 cfu/100 ml based upon a statistical threshold value of the 95th percentile.” Lines 69-73 in revised manuscript (word version, track changes on).
In introduction, authors need to highlight the novelty and importance of this study with respect to the existing findings in the scientific literature.
Thank you for your comment. We added the following: “We should point out that Kifisos river is the main urban river of the capital city of Greece (Athens) which nevertheless is not monitored adequately regarding water pollution and therefore the relevant information is scarce. Thus, this effort is probably the first one attempting to present a recent spatiotemporal analysis for the water quality status of the river, by using several variables and attempting to identify pollution hot-spots along its course.” Lines 88-92.
“This is a first attempt to investigate correlations between E. coli and HWQI in an urban river, which is crucial for future water quality assessments.” Lines 103-105, in revised manuscript (word version, track changes on).
Line 128: “five sampling sites (K Ekv, K 5, K 12, K 17, K MD).” Could you explain on which basis these five water sampling sites were selected. Was the sampling done on the surface water or certain deep from the surface.
“Five sampling locations were selected that are distributed along the drainage catchment from its upstream part where limited pollution sources exist to the river mouth (Fig. 1). Monthly water surface collection was conducted in between 10:00 am and 1:00 pm, from May 2021 to April 2022 from five sampling sites (K Ekv, K 5, K 12, K 17, K MD).” Lines 135-144, in revised manuscript (word version, track changes on).
The depth of water during most sampling effort was between 20 and 40 cm and therefore the samples were collected from the surface.
Line 132: “HgCl2,” Check the typo error here. Clearly mention the role of HgCl2, and whether addition of HgCl2 interferes the measurement of the water quality parameters by analytical methods.
HgCl2 is used as a preservative for the nutrients’ samples only and is a standard procedure to stop biological activity in the collected water which otherwise could alter the actual concentration from the sampling moment until the laboratory analysis. Thus, the addition of the specific preservative does not affect negatively the chemical analysis of the nutrient samples.
Line 259 – 260: “which is within the expected range for freshwater” Show the values for the concentration range here.
The expected range for freshwater is from 100 to 1000 μS/cm. To improve the understanding of this part in the manuscript we modified the phrase. The highest values of EC in most of all the other cases were close to 1000 μS/cm. This value is within the expected range for freshwater in an urban river. Lines 286-288, in revised manuscript (word version, track changes on).
Line 308 – 312: Strengthen the discussion on the correlation between physicochemical water quality parameters and microbial (E. coli) water quality, i.e., which of the physicochemical water quality parameters are strongly correlated with E. coli level in each sampling site.
We thank the reviewer for this comment. The results and the discussion section have been enhanced in the revised manuscript according to the suggestions of the reviewer.
Results section: “Moreover, Pearson correlation coefficient (r) per site was calculated (Table 4) to examine the patterns between E. Coli and the physicochemical parameters (T, EC, pH and DO). A strong negative correlation was estimated between EC and E. coli for two (K 12 and K MD) out of five sites. For these two sites EC and E. coli concentrations linear regression models were developed. These models resulted a coefficient of determination R-squared (R2) of 0.63 for K MD site and a R2 of 0.15 for K 12.” Lines: 322-327.
Discussion section: “EC can indirectly influence the presence and behavior of E. coli since it is often associated with the concentration of dissolved ions in water, including nutrients like phosphates and nitrates, which may support the growth and survival of E. coli bacteria. These contaminants could originate from various sources, such as industrial discharges or agricultural runoff, and can affect the viability of E. coli populations. However, EC is not a direct measure of E. coli presence or concentration. A coefficient of determination R-squared (R2) of 0.63 for K MD site indicated that the two variables are likely related with each other. However, more data are necessary for decisive validation of the relationships developed in this study.” Lines: 322-327.
Table 1: The heading (a/a) for the first column is not clear. Change the heading of the second column to “Parameter (Unit)”. Also, in the footnote, mention about the meaning of the bolded texts in the table.
Done.
Round 2
Reviewer 1 Report
This manuscript reports on the water quality of the Kifisos river in Greece from its headwaters to its inlet at five stations during different months of the year. Although the data provide some feedback, the overall novelty is lacking and many of the results and impacts are predictable. If the author still wants to publish this manuscript, I have some suggestions:
- Please provide all raw data of different parameters for each month in the form of a supplementary file.
We created a new supplementary file (Table S1) including the data supporting the findings of this study. Line 221 in revised manuscript (word version, track changes on).
Done
- Please double check the appropriateness of the cited paper, for example, lines 75 to 77, I have checked the cited paper and there is no mention of volcanic eruptions and agricultural fires as the main natural causes, please read the cited paper again carefully.
We double checked the cited paper of Manoli & Samara, (1999). More specifically it is mentioned in the introduction section that: “Volcanic eruptions and forest and prairie fires are among the major natural sources of PAHs in the atmosphere.”
You mean “agricultural fires”and “forest and prairie fires”mean the same thing? ; Please still indicate that this is themain source in the atmosphere.
- Please improve the clarity and quality of the figures in the manuscript, such as figure 2 and figure 5.
We improved the clarity and quality of the specified figures 2 and 5.
Done
- Some microbial generic names are not in italics, such as line 58, and some punctuation mark is abused. Please check the manuscript again.
Microbial generic names have been replaced with italics throughout the revised manuscript. We also fixed the punctuation marks.
Done
- In the discussion section, it is necessary to compare the research results of this manuscript with those of published articles and explain the reasons for similarities or differences. The conclusion section should summarize the specific results and unique findings of this manuscript, rather than describing some common sense such as "Urban diffuse pollution from cities and transport, carries pollutants from roads and urban surfaces into storm drains and out into watercourses at many locations along the river."
The reviewer is right and, thus, we moved the respective sentence to introduction. Lines 34-36 in revised manuscript (word version, track changes on).
Unfortunately, there are no many studies conducted in the past in Kifisos river and therefore the comparison in the discussion section could only be done with limited relevant research efforts from the international literature (eg. Argyraki Koutsomitrou et al., 2014, Argyraki et al. 2013, Adeniji, Okoh, et al., 2019; Markogianni, Anastasopoulou, et al., 2018, Panagopoulos, Alexakis, et al., 2022).
Comparative discussions are not just about simply comparing studies on this one river, but other similar river studies can be compared. The purpose of the comparison is to be used to discover the uniqueness and novelty of your study and thus mention the attractiveness of the manuscript, on which the author may not have spent enough time and effort.
Author Response
- Responses provided in blue text –
We appreciate the opportunity to revise and resubmit our manuscript. We appreciate the time and effort that the reviewer has dedicated to providing his/her valuable feedback giving us the opportunity to improve our manuscript.
You mean “agricultural fires”and “forest and prairie fires”mean the same thing? ; Please still indicate that this is themain source in the atmosphere.
As highlighted by the reviewer, the phrasing we used on this was ambiguous and therefore we rephrased the sentence in order to distinct between the two aforementioned sources. Furthermore, the word “main” was removed.
“Natural sources include volcanic eruptions and forest fires while incomplete combustion of transportation fuel, industrial waste, agricultural fires and waste incineration constitute examples of anthropogenic sources (Manoli & Samara, 1999; Zhang & Tao, 2009).”
Zhang, Y., & Tao, S. (2009). Global atmospheric emission inventory of polycyclic aromatic hydrocarbons (PAHs) for 2004. Atmospheric Environment, 43(4), 812–819. https://doi.org/10.1016/j.atmosenv.2008.10.050

Reviewer 2 Report
The authors have answers the queries satisfactorily and the manuscript can be published.
Author Response
Thank you.
Reviewer 3 Report
The author's rebuttal to most of the comments is appropriate. Still some places in the manuscript (Line 16, 323, 329, etc.), it is written as "E. Coli" instead of "E. coli" that I pointed out previously. The manuscript can be accepted after the above minor correction.
Minor typographical/grammatical related errors correction is required to the manuscript.
Author Response
- Responses provided in blue text –
We appreciate the time and effort that the reviewer has dedicated to providing his/her valuable feedback giving us the opportunity to improve our manuscript. We appreciate the opportunity to revise and resubmit our manuscript.
The author's rebuttal to most of the comments is appropriate. Still some places in the manuscript (Line 16, 323, 329, etc.), it is written as "E. Coli" instead of "E. coli" that I pointed out previously. The manuscript can be accepted after the above minor correction.
Thank you for your comment. We corrected the "E. Coli" to “E. coli” throughout the revised manuscript.
Minor typographical/grammatical related errors correction is required to the manuscript.
The manuscript has been edited by an English-speaking native, so we hope it now matches the journal standard.